# Fixed (Trackside) Energy Storage System for DC Electric Railways Based on Full-SiC Isolated DC-DC Converters

**Joseph Fabre** [1,2,*], **Philippe Ladoux** [2,*] **and Hervé Caron** [3,*]

1 SCLE-SFE, 25 Chemin de Paléficat, 31204 Toulouse, France
2 Laboratory of Plasma and Energy Conversion (LAPLACE), Université de Toulouse, 31000 Toulouse, France
3 Département Traction Électrique, SNCF Réseau, 93418 La Plaine Saint-Denis, France
* Correspondence: joseph.fabre@scle.fr (J.F.); philippe.ladoux@laplace.univ-tlse.fr (P.L.);
herve.caron@reseau.sncf.fr (H.C.)

**Abstract:** At present, in several European railway networks using traditional DC electrification systems, it is not possible to increase traffic nor to operate locomotives at their nominal power ratings. Trackside energy storage systems (TESSs) can be an alternative solution for the creation of new substations. A TESS limits contact line voltage drops and smooths the power absorbed during peak traffic. Thus, the efficiency of the power system can be increased while limiting costs and the environmental impact. This paper proposes a new topology of a TESS based on full-SiC isolated DC/DC converters associated with lithium-ion batteries and galvanic isolation, offering major advantages for operational safety. In the event of a fault, the input and output terminals of the converters are electrically separated, and the contact line voltage can never be directly applied to the batteries. In addition, the use of SiC MOSFETs makes it possible to obtain excellent efficiency with a high switching frequency. The first part of this paper presents the main characteristics of an elementary TESS module, while the second part proposes a sizing methodology for the typical case of a 1.5 kV DC line, which shows the limits of using TESSs to reinforce a power supply. Finally, the experimental results of an elementary module prototype are presented.

**Keywords:** energy storage system; railways; DC-DC power converters; power MOSFET; silicon carbide; R-SAB converter

## 1. Introduction

On DC-electrified lines, an increase in traffic, both for freight and passenger mobility, requires, in certain sectors, the installation of additional substations to maintain the contact line voltage within an acceptable range. Nevertheless, adding a new substation often requires the creation of a dedicated power transmission line. Thus, such a project can be extremely expensive and the implementation of a trackside energy storage system (TESS) [1], based on batteries, can be an alternative solution (Figure 1). The efficiency of the power system can be increased while limiting costs and the environmental impact. Regarding the connection of batteries to the contact line, an isolated converter topology provides a major advantage for operational safety. In the event of internal faults linked to gate driver or semiconductor failures, the contact line voltage can never be directly applied to the battery terminals.

Until now, no TESS has been implemented on the French rail network, and there is no dedicated industrial product, at least not in France nor Europe. Japan Railways are the pioneers in this field and have installed around 20 battery storage units of several 100 kWh for more than 10 years on their 1.5 kV DC lines [1,2]. The power electronics solutions developed by Japanese manufacturers do not provide galvanic isolation between the traction circuit and the batteries. Thus, in the event of an internal converter fault, the contact line voltage could be applied to the terminals of the battery, causing an explosion and/or a fire.

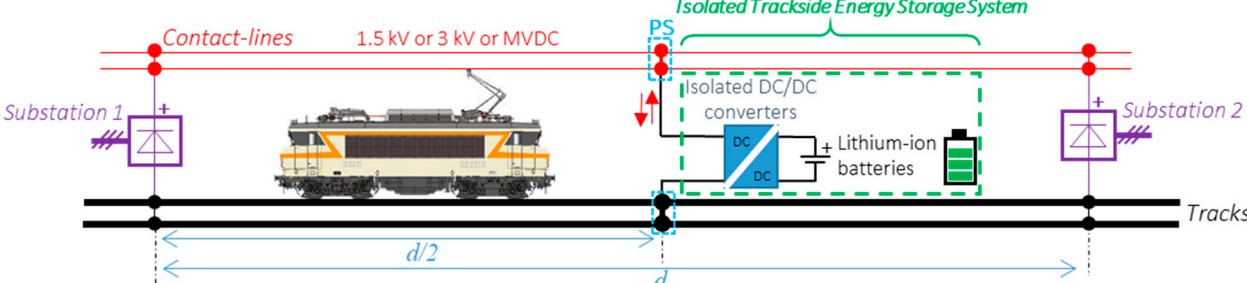

**Figure 1.** Principle of reinforcement of a DC railway line sector with isolated trackside energy storage system (TESS) installed in the middle of the sector (at *d/2*) at the paralleling station (PS).

To overcome this major drawback, "SNCF Innovation et Recherche" and "SNCF-Réseau" have collaborated with LAPLACE since 2019 to study a very efficient and compact isolated converter at the MW scale. The availability of SiC-MOSFET modules with voltage and current ratings compatible with railway applications and the development of magnetic materials capable of operating at a high frequency (up to several tens of kHz) have paved the way for the realization of a prototype of an elementary isolated DC/DC converter rated at 300 kW and operating at a voltage of 1800 V with an efficiency close to 99% [3,4]. For safety and lifespan reasons, a lithium ferro phosphate battery was selected and associated with this isolated DC/DC converter to form an elementary module rated at 300 kW/147 kWh.

## 2. Prototype of the Elementary Module of the TESS

The isolated DC/DC converter is composed of a resonant dual-bridge converter (RDBC), including a medium-frequency transformer (MFT) [3–5], cascaded with a three-level chopper (Figure 2). Even though this solution requires voltage control of the capacitive midpoint [6], a 3-level chopper topology was chosen because of the availability of 1700 V SiC power modules already in stock at the laboratory. The RDBC and the MFT provide galvanic isolation, while the chopper regulates the battery current. In the industrial version, the chopper is based on 3.3 kV/750A SiC power modules. Two interleaved chopper legs are connected in parallel with coupled inductors (intercell transformer (ICT)), which will allow an appreciable gain in volume [7–10].

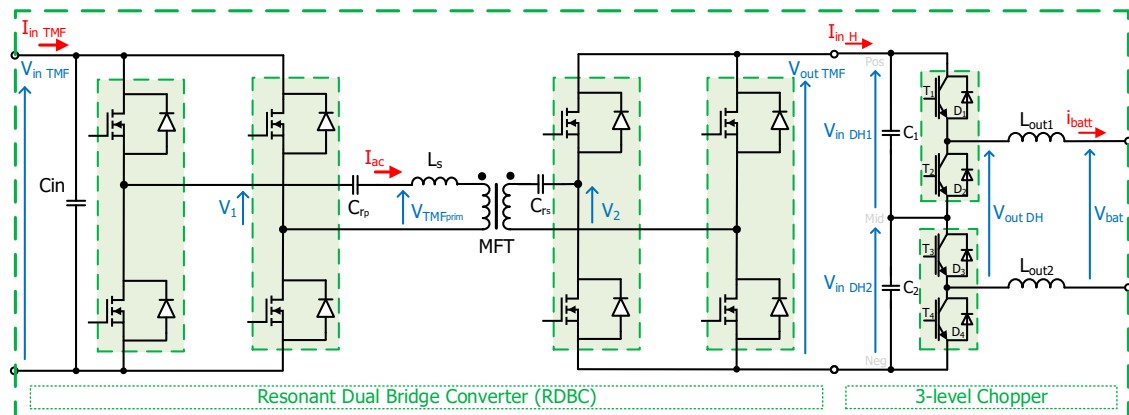

**Figure 2.** Isolated DC/DC converter: principle of an elementary block of the current prototype.

A lithium ferro phosphate (LFP) battery technology was selected [11] and associated with a battery management system (BMS); LFP is currently one of the most secure lithium-ion technologies on the market [12]. It exhibits long-term voltage and current stability and offers a longer life than other lithium-ion chemistries [13]. Despite having a lower energy density than other lithium chemistries, LFP batteries are currently used in several applications such as electric vehicles and uninterruptible power supplies. Tables 1 and 2

summarize the characteristics of the power converter and its MFT. The sizing and the choice of components were already presented in papers [3,4]. The gate-driver of the SiC-MOSFETs was designed at the LAPLACE laboratory [14]. The output inductances $L_{out1} = L_{out2}$ must be at least 282 µH to ensure a current ripple of less than 5% of the maximum battery current. In addition, in this prototype, 2 air-core inductors available in the test platform of 1 mH each were used during the tests. Table 3 presents the characteristics of the selected LFP battery.

**Table 1.** Isolated DC/DC converter prototype: RDBC DCM and 3-level chopper specifications.

|  | **RDBC DCM** | **3-Level Chopper** |
|---|---|---|
| Input voltage | $V_{in}$ = 1800 V | $V_{in}$ = 1800 V |
| Converter nominal output current | $I_{out}$ = 170 A | $I_{out}$ = 400 A |
| Converter nominal output power | $P_{out} \approx$ 300 kW | $P_{out} \approx$ 300 kW |
| SiC-MOSFET HBM | Mitsubishi 3.3 kV/750 A | ABB 1.7 kV/1100 A (custom power module) |
| Gate driver (with optical interface) | $V_{gs}$ = −5/+17 V | $V_{gs}$ = −5/+20 V |
| Switching frequency | $f_{sw}$ = 15 kHz | $f_{sw}$ = 10 kHz |
| Resonance frequency | $f_0$ = 19.37 kHz |  |
| Resonant capacitor | Illinois Capacitor; HC5 series Series capacitor: $C_{rp} = C_{rs}$ = 6 µF |  |
| Output inductances |  | $L_{out1} = L_{out2}$ = 282 µH (5% maximum current ripple) $L_{out1} = L_{out2}$ = 1 mH (prototype) |
| Dead-time on voltage source inverter | 8 µs | 3 µs |
| Semiconductor heatsinks | Water cold plates from Mersen | |
| Water flow rate | $Q_v$ = [1 to 5] L/min | |
| Input/output capacitors | Electronics Concepts; LH3 series | |
| Sensors | LEM series LV and DV | |
| Control system | Imperix BoomBox with optical interface | |

**Table 2.** Isolated DC/DC converter prototype: MFT specifications [3].

| | |
|---|---|
| Input voltage | $V_{in}$ = 1800 V |
| Turns ratio | 1:1 |
| Nominal current | $I_{rms}$ = 223 A |
| Apparent power | $S_{rms}$ = 400 kVA |
| Isolation level | $V_{iso}$ = 20 kV (tested @5 kV) |
| Total weight | 160 kg |
| Volume | 45 L |
| Cooling fluid | Water (flow rate of 5 L/min) |
| Magnetizing inductance | $L_m$ = 5.7 mH |
| Total leakage inductance | $L_s$ = 22.5 µH |
| Total equivalent series resistances | $R_s$ = 17.5 mΩ |
| Core loss equivalent resistance | $R_m$ = 7.9 kΩ |

**Table 3.** Specifications of the lithium iron phosphate battery.

| | |
|---|---|
| Nominal voltage | $V_{\text{batt-nom}}$ = 614 V |
| Voltage range | $V_{\text{batt}}$ = [500 V to 710 V] |
| Energy (C/5, 23 °C) | $E_n$ = 147 kWh |
| Capacity (C/5, 23 °C) | $Q_n$ = 240 Ah |
| Weight | 1200 kg |
| Max. discharge current | $I_{\text{discharge-max}}$ = 400 A |
| Max. charge current | $I_{\text{charge-max}}$ = 100 A |
| Composition | 2 parallel legs each with 12 elementary racks in series |
| Cooling system | Air |

In the final application, in order to allow for the correct power to be supplied to a railway line sector, *N* isolated DC/DC converters are associated in parallel, as shown in Figure 3. This modularity allows the mass production of elementary converters, thus decreasing the manufacturing cost while facilitating assembly and maintenance.

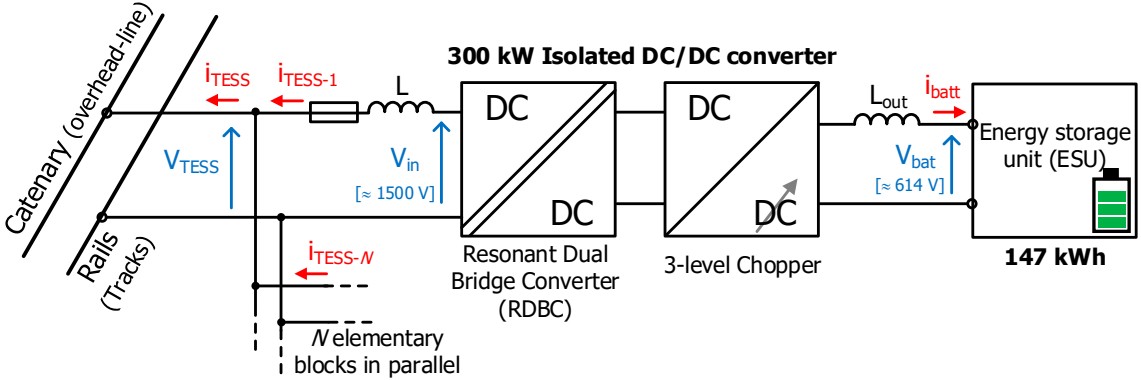

**Figure 3.** Paralleling principle for reinforcing a DC railway line sector with *N* elementary isolated trackside energy storage systems.

## 3. Circuit Modelling and Sizing at the System Scale

### 3.1. Battery Charge/Discharge Control Method and Isolated DC/DC Converter Modeling

As shown in Figure 4a, the control is based on two cascaded loops. Figure 4b illustrates the operation of the TESS connected at the PS. The external loop regulates the contact line voltage $V_{\text{TESS}}$ to $V_{\text{threshold}}$ as long as the battery current limits are not reached. Otherwise, when a train approaches the TESS, the output voltage $V_{\text{TESS}}$ decreases and the battery is discharged with the current $I_{\text{discharge\_max}}$. Likewise, when trains are away from the TESS or when there is no traffic, the output voltage $V_{\text{TESS}}$ increases, and the battery is recharged with the current $I_{\text{charge\_max}}$. The two current limits are fixed by the battery management system (BMS) according to various criteria such as the state of charge (SoC) or the temperature of the cells.

The sizing of the TESS for a given line sector requires that the isolated DC/DC converter be modeled using a simple averaged model, as shown in Figure 5. In addition, as a first approximation, the battery can be modeled using a DC voltage source with an internal resistance [15,16]. Then, for *N* converters in parallel, a vectorized model is implemented in PLECS$^{\text{TM}}$ software [17]. To reduce the simulation time for railroad traffic lasting for several hours, an averaged model of the converter and its control is implemented.

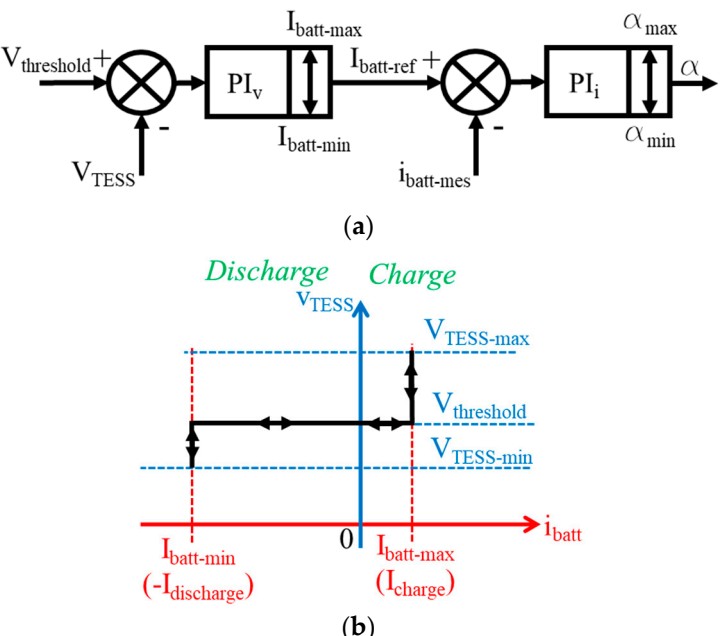

**Figure 4.** Control of DC/DC converter based on two cascaded loops: (**a**) voltage and current control loops; (**b**) voltage ($V_{TESS}$) – current ($i_{batt}$) characteristic.

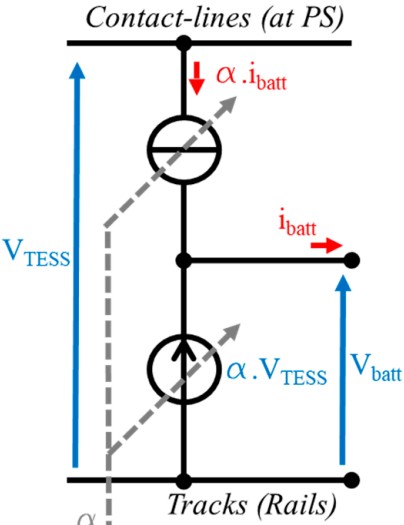

**Figure 5.** DC/DC converter averaged model.

### 3.2. Modeling of a Double-Track Sector with Symmetrical Circulation of Trains

For several years, freight and regional passenger rail traffic have increased steadily. As a result, more and more electrical power is required, which causes important losses in the overhead line as well as major voltage drops. Thus, in some sectors, locomotives cannot run at their nominal power, which limits traffic. In order to limit overhead-line voltage drops and increase the power to a sector, it is necessary to reduce the electrical circuit resistance. Two solutions are usually combined [18]: a feed wire is installed in parallel to the contact line, and a paralleling connection of the electrical circuits of the tracks is implemented at the sector midpoint (paralleling station (PS)) (Figure 6).

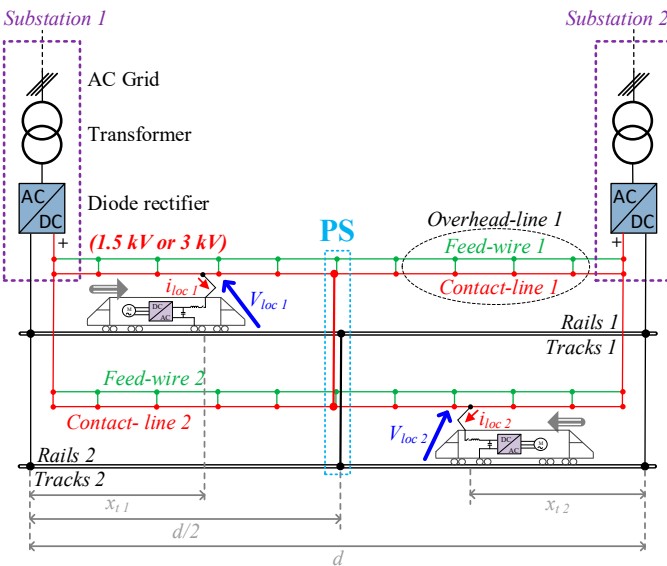

**Figure 6.** Classical DC railway electrification system: double-track line with paralleling station (PS) and contact line with parallel feed wire (overhead line).

For reinforcement purposes, installing a new substation at the paralleling station is the classic way to reduce voltage drops (Figure 7a). However, such a solution may be limited by environmental and financial constraints. To overcome this problem, a solution based on a three-wire supply system has been proposed in [18,19]. Nevertheless, a new feed wire, operating at a medium voltage level, must be installed on the overhead line poles, which can also be expensive. Therefore, the installation of a TESS, presented in Figure 7b, can be a competitive solution.

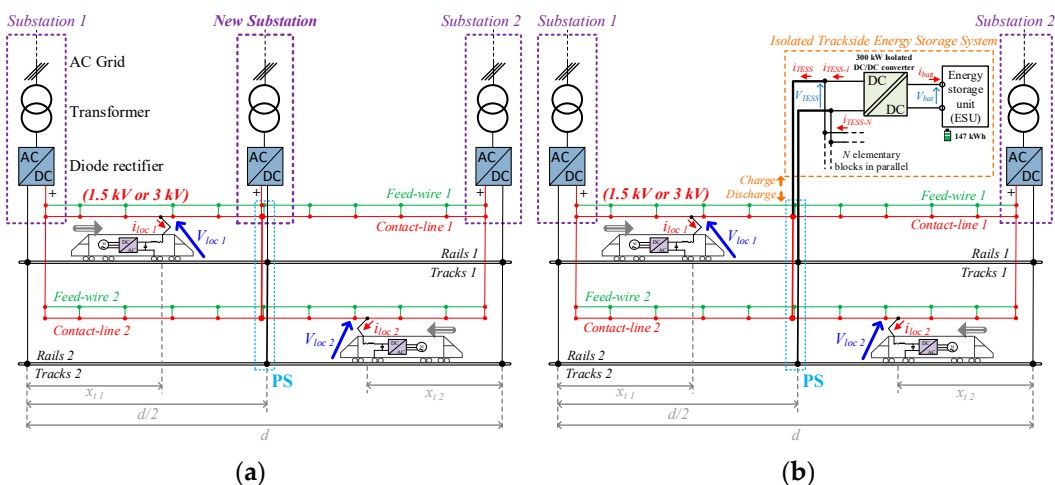

(**a**)                         (**b**)

**Figure 7.** Solutions to reduce voltage drops: (**a**) double-track line equipped with a new substation at PS; (**b**) double-track line equipped with an isolated trackside energy storage system at PS.

In order to obtain a straightforward analysis, it was decided to implement the electrical circuit in *PLECS$^{TM}$* software. With the time-constant of the electrical circuit being negligible in comparison with train dynamics, a steady-state DC operation was considered. The substations are assumed to behave as DC voltage sources with an internal resistance $R_{int}$ and an electromotive source $E_0$. Each train is modeled with a current source. The rails and the overhead line are represented only by resistances, and the positions of the trains are modeled via resistance variations.

This choice makes it possible to study double-track lines with the circulation of trains while considering the behavior of the isolated DC/DC converter, the battery, and the control system. The electrical parameters of the circuit are listed in Table 4.

**Table 4.** Electrical parameters for a typical double-track line.

| | |
|---|---|
| Rectifier unloaded voltage | $E_0$ = 1750 V |
| Rectifier internal impedance including transformer reactance | $R_{int}$ = 25 m$\Omega$ |
| Overhead line cross-section for each line (copper) | $S$ = 630 mm$^2$ |
| Rail size (linear weight of the rail) | $w$ = 60 kg/m |
| Distance between two substations | $d$ = 15 km |
| Overhead line resistance per kilometer | $R_{cl}$ = [18.8/$S$] $\Omega$/km |
| Rail resistance per kilometer (per each track) * | $R_{rl}$ = [0.9/$w$] $\Omega$/km |

* The two tracks are electrically in parallel. They behave as a single conductor.

### 3.3. Power Sizing of the TESS—Simulations with Two Trains in the Sector

The European Standard EN 50163 [20] requires a minimum pantograph voltage of 1 kV for a 1.5 kV electrification system. Nevertheless, it should be noted that at this voltage level, a modern locomotive cannot operate at its nominal power. In fact, the control system of the locomotive limits the absorbed power when the pantograph voltage is lower than 1300 V. As a result, the trains cannot keep to schedule, and the traffic on the railway line is disturbed. Thus, the power sizing of a TESS must be determined with the view of always ensuring a sufficient pantograph voltage.

Assuming that the two trains, running in opposite directions in the sector, operate at the same power level, the minimum value of the pantograph voltage is determined only by considering the crossing point of the trains [21]. The simulation results are presented in Figure 8. They show that without the TESS, the pantograph voltage of the locomotive will be lower than 1300 V if the power absorbed by each locomotive is greater than 3 MW. For locomotives with a nominal power of 4.5 MW, the minimum value of the pantograph voltage decreases to 1081 V despite the fact that the on-board control system limits the current drawn from the overhead line.

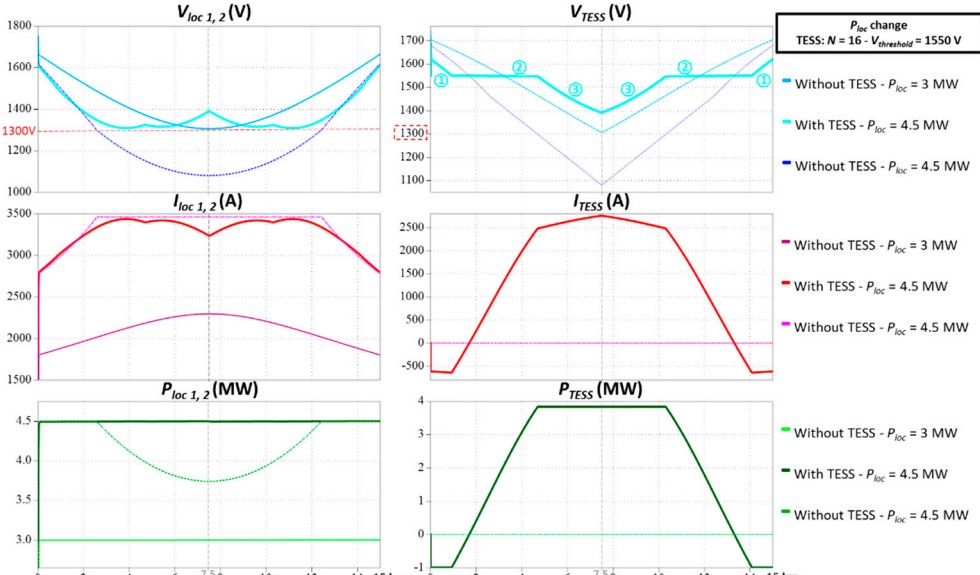

**Figure 8.** Simulation results according to the location of the crossing point of locomotives 1 and 2 in the sector. A total of 3 case studies: $P_{loc}$ = 3 MW—without TESS, and $P_{loc}$ = 4.5 MW—with and without TESS.

In order to guarantee a pantograph voltage higher than 1300 V, and taking into account the battery specified in Table 3, the TESS must include 16 elementary blocks in parallel. In Figure 8, the plot of $V_{TESS}$ highlights the 3 operating areas previously presented in Figure 4b: *Zone 1* corresponds to the battery charge ($V_{TESS} > V_{threshold}$); *Zone 2* corresponds to a voltage source operation ($V_{TESS} = V_{threshold} = 1550$ V); and *Zone 3* corresponds to battery discharge at its power limit ($V_{TESS} < V_{threshold}$).

### 3.3.1. Choice of the Charge/Discharge Threshold Voltage ($V_{threshold}$)

Figure 9 illustrates the influence of the threshold voltage. Thus, if $V_{threshold} = 1500$ V, the pantograph voltage $V_{loc\ 1,2}$ drops below 1300 V, and the current absorbed by the trains is then limited. On the other hand, if $V_{threshold} = 1600$ V, the pantograph voltage $V_{loc\ 1,2}$ is always higher than 1300 V. In this case, the minimum value of $V_{loc\ 1,2}$ no longer depends on $V_{threshold}$ but on the current injected by the TESS on the overhead line. Finally, if $V_{threshold}$ is set to 1550 V, the minimum pantograph voltage $V_{loc\ 1,2}$ is just over 1300 V, and the plot of $I_{TESS}$ shows that this also offers a trade-off regarding the charge range of the battery.

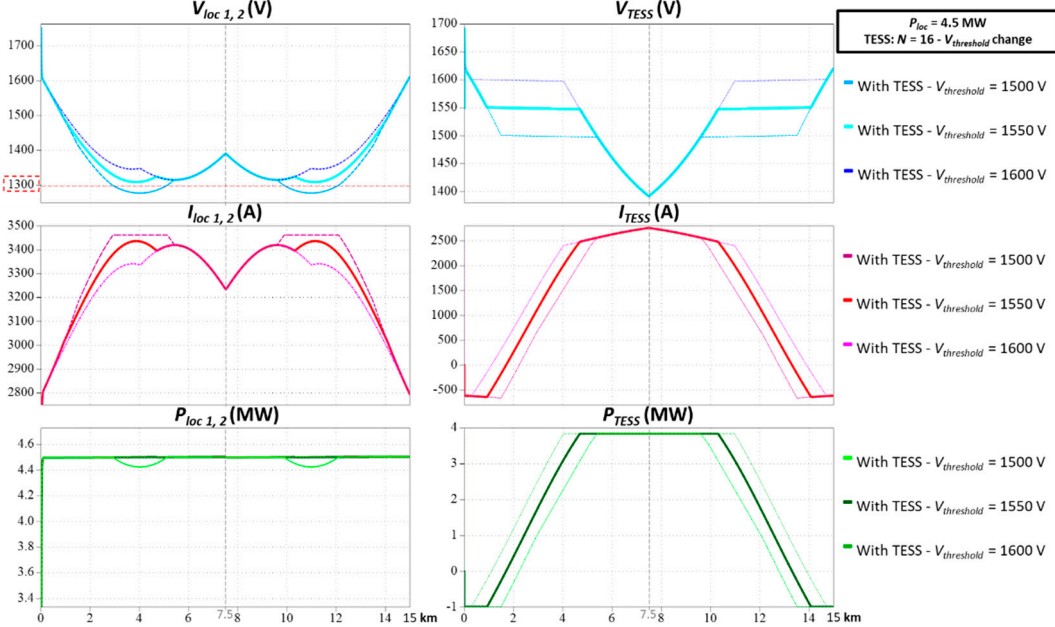

**Figure 9.** Simulation results according to the location of the crossing point of locomotives 1 and 2 in the sector. Analysis of the influence of $V_{threshold}$ ($N = 16$; $P_{loc} = 4.5$ MW).

### 3.3.2. Influence of the Choice of the Number of Elementary Blocks in Parallel ($N$)

Figure 10 illustrates the influence of the number of elementary blocks in parallel ($N$). A total of three different values are compared: $N = 14$, 16, and 18 for $V_{threshold}$ at 1550 V. Thus, if $N < 16$, the pantograph voltage $V_{loc\ 1,2}$ drops below 1300 V, and the current absorbed by the trains is then limited. When $N = 16$, the minimum pantograph voltage $V_{loc\ 1,2}$ is just over 1300 V. For $N > 16$, the minimum value of pantograph voltage $V_{loc\ 1,2}$ depends only on the parameter $V_{threshold}$.

The simulation results presented in Figures 9 and 10 show that there is an optimal choice of the couple $N$ and $V_{threshold}$. This couple must be determined to ensure an acceptable pantograph voltage (generally above 1300 V) allowing for the specificities of the railway line sector (the substation spacing, overhead line cross-section, and weight of a rail per unit of length) and the power of the trains in circulation.

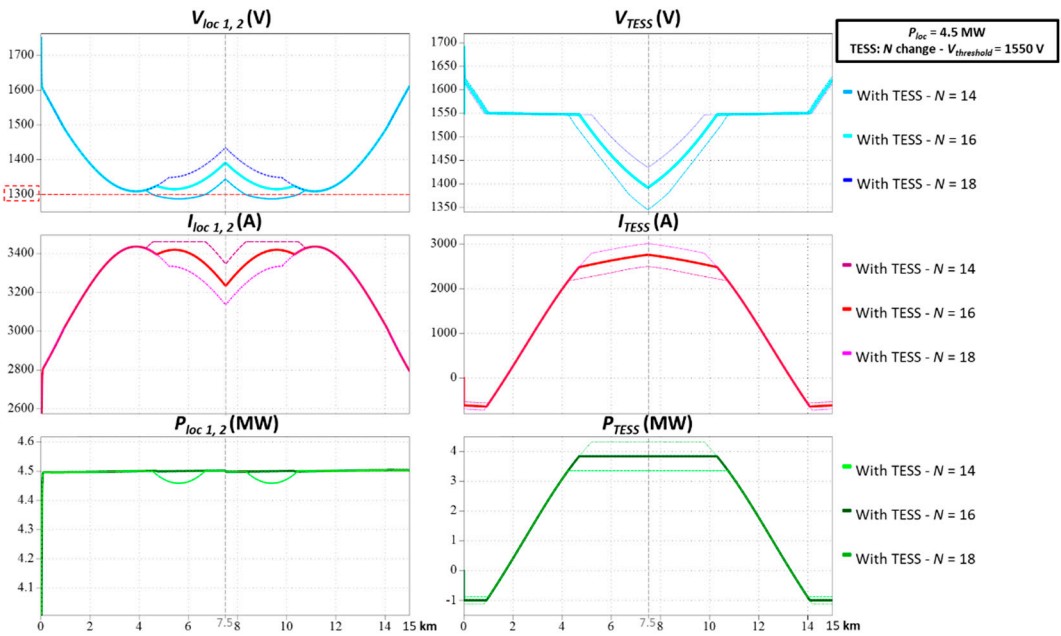

**Figure 10.** Simulation results according to the location of the crossing point of locomotives 1 and 2 in the sector. Analysis of the influence of $N$ ($V_{threshold}$ = 1550 V; $P_{loc}$ = 4.5 MW).

### 3.3.3. Maximum Train Power in a Sector

On the basis of the previous circuit model, considering only the train-crossing location (which is the worst-case location), Figure 11 shows the minimum voltage at the pantograph of the locomotives ($V_{loc-min}$) according to their absorbed power ($P_{loc}$). The calculation results are presented in the initial conditions of the sector, with the TESS ($N$ = 16 and $V_{threshold}$ = 1550 V) and with a new substation ($E_0$ = 1750 V; $R_{int}$ = 25 mΩ) installed at the paralleling station (the sector midpoint). In a modern locomotive, the on-board control system limits the current absorbed if the pantograph voltage drops below 1300 V. Thus, below this value, the absorbed power can still be slightly increased down to 1100 V, which is the minimum operating voltage allowed by the standard EN50163 [20].

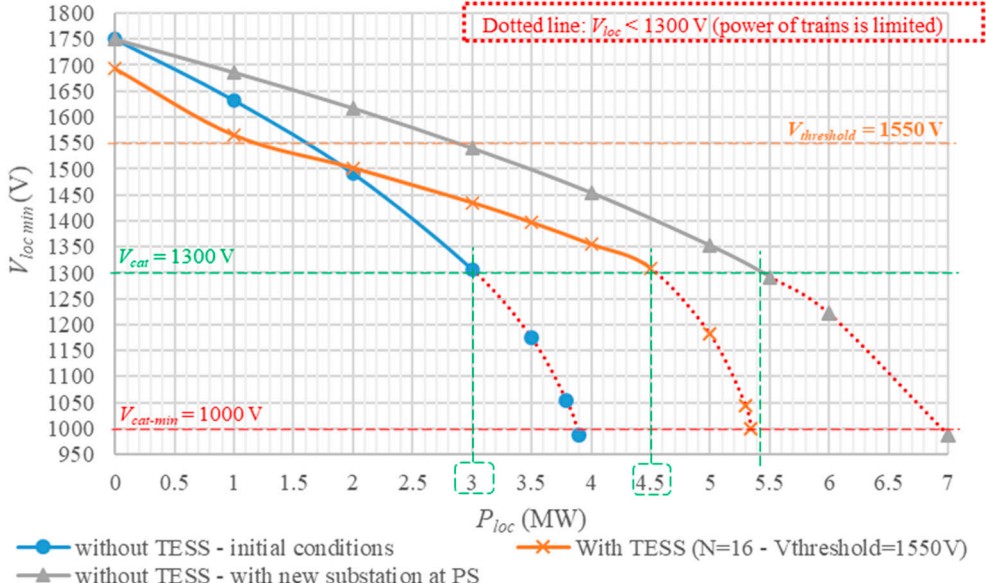

**Figure 11.** Minimum voltage at the terminals of the locomotives according to their power—with and without TESS ($N$ = 16 – $V_{threshold}$ = 1550 V)—and with a new substation ($E_0$ = 1750 V; $R_{int}$ = 25 mΩ) at PS, position $d/2$.

In the initial conditions (without the TESS), the total power available at the crossing of the trains in the middle of the section is 6 MW, or 3 MW per locomotive. In the configuration with the TESS, it is possible to reach 9 MW, or 4.5 MW per locomotive. Meanwhile, when adding a new substation at PS, position $d/2$, it is possible to reach 10.8 MW, or 5.4 MW per locomotive.

### 3.3.4. The Limits of the TESS Installation

Sections 3.3.1–3.3.3 clearly showed that the installation of a TESS can be an alternative to the creation of a new substation. Considering the characteristics of the battery and the typical parameters of a line sector, given in Tables 3 and 4, respectively, Table 5 presents, for different train powers, the number of elementary blocks required to guarantee a pantograph voltage above 1300 V, whatever the crossing point of the trains in the sector. Increasing $N$ beyond 22 is equivalent to installing a new substation. Only a complete cost analysis would make it possible to choose this solution. This has to be carried out on a case-by-case basis depending on the geographical location of the sector and the possibilities of connecting a substation to the three-phase public network.

**Table 5.** Number of elementary blocks required to maintain a minimum pantograph voltage at 1300 V versus the power of the trains.

| $P_{loc}$ (MW) | TESS Implemented at PS (Position $d/2$) | $N$ | $V_{threshold}$ (V) |
|---|---|---|---|
| 3 | Initial conditions | 0 | x |
| 3.5 | TESS | 5 | 1430 |
| 4 | TESS | 10 | 1480 |
| 4.5 | TESS | 16 | 1550 |
| 5 | TESS | 22 | 1610 |
| 5.4 | New substation ($E_0$ = 1750 V; $R_{int}$ = 25 mΩ) | 0 | x |

### 3.4. Modeling of a Line Sector with Representative Railroad Traffic

In the previous sections, the analysis of the voltage at the pantograph of the locomotives was carried out considering the worst case corresponding to the crossing of two trains. This allowed the sizing of the TESS in terms of power. Thus, to evaluate the energy to be provided, it is necessary to consider the railroad traffic in the sector.

The case with $P_{loc}$ at 4.5 MW is considered, and according to the analysis presented in the previous sections, the TESS connected at PS is composed of 16 elementary units in parallel, and the total stored energy is 16 × 147 kWh = 2.352 MWh.

The target in terms of railroad traffic is 1 train every 10 min left to right (odd trains) and 1 train every 11 min right to left (even trains). Trains run at a constant speed of 120 km/h (taking 7.5 min to travel the sector). Therefore, the train spacing is 20 km for odd trains and 22 km for even trains. As a result, in a sector with a length of 15 km, there are only 2 trains running in opposite directions. In the initial conditions, if the pantograph voltage is lower than 1300 V, the power of the trains is reduced. Figure 12 shows the railroad traffic corresponding to the rush hours with the circulation of 23 trains in the sector over 120 min. The crossing points of the trains are represented by black points.

The simulations were performed with *PLECS*$^{TM}$ software considering the control circuit and the averaged model of the DC/DC converter presented in Figures 4 and 5, respectively. The electrical characteristics of the double-track line sector are those detailed in Table 4.

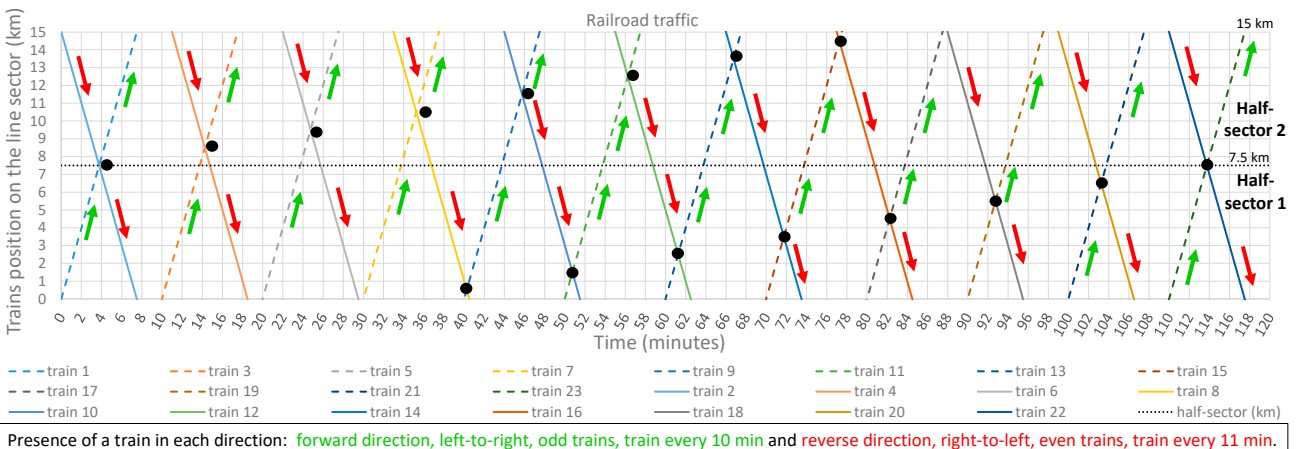

**Figure 12.** Railroad traffic considering the circulation of 23 trains over a time span of 2 h.

### 3.5. Analysis of the Simulation Results

Figure 13 shows the simulation results in the initial conditions with the TESS installed. In the initial conditions, when the trains cross in the middle of the sector (at PS), the pantograph voltage drops to 1100 V. This means that the efficiency of the traction system is seriously degraded. When the TESS is installed, the maximum power delivered per storage unit is 240 kW, giving a total power of 3.840 MW. During the 2 h of peak traffic, the TESS must provide 1.921 MWh, which means that the depth of discharge reaches 81.7%. Therefore, in view of the current limitation during charging (100 A per storage unit), an off-peak period of 2 hours without any train traffic is necessary to fully recharge the batteries.

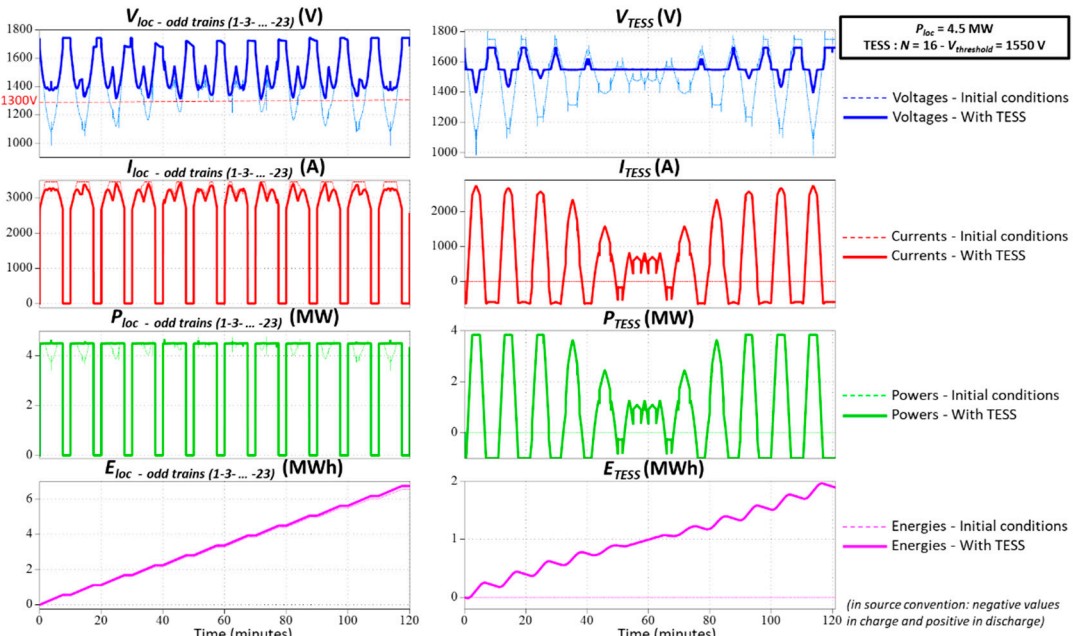

**Figure 13.** Simulation results with the circulation of 23 trains over a time span of 2 h (peak period); initial conditions and with TESS.

Table 6 shows the energy balance for different configurations of the line sector (with the initial conditions, reinforcement with the TESS, and installation of a new substation).

According to these results, the efficiency of the power system can be calculated according to (1).

$$\eta = \frac{\sum E_{loc}}{\sum E_{substation}} \tag{1}$$

**Table 6.** Energy balance of the double-track line sector.

| | Peak Period | | | Off-Peak Period (without Train Traffic) |
|---|---|---|---|---|
| | Initial Conditions | With TESS (*N* = 16) | With a New Substation | With TESS (*N* = 16) |
| Substations | | | | |
| $E_{substation\ 1}$ (MWh) | 7.769 MWh | 6.465 MWh | 3.924 MWh | 0.989 MWh |
| $E_{substation\ 2}$ (MWh) | 7.769 MWh | 6.465 MWh | 3.924 MWh | 0.989 MWh |
| $E_{new\ substation\ at\ PS}$ (MWh) | x | x | 6.351 MWh | x |
| $\sum E_{substation}$ (MWh) | 15.538 MWh | 12.930 MWh | 14.199 MWh | 1.978 MWh |
| Trains | | | | |
| $E_{loc\ -\ odd\ trains\ (1\text{-}3\text{-}\ldots\ \text{-}23)}$ (MWh) | 6.563 MWh | 6.749 MWh | 6.749 MWh | 0 MWh |
| $E_{loc\ -\ even\ trains\ (2\text{-}4\text{-}\ldots\ 22)}$ (MWh) | 5.998 MWh | 6.186 MWh | 6.186 MWh | 0 MWh |
| $\sum E_{loc}$ (MWh) | 12.561 MWh | 12.936 MWh | 12.936 MWh | 0 MWh |
| TESS | | | | |
| $E_{TESS}$ (MWh) | x | 1.921 MWh | x | −1.921 MWh |

Compared with the initial conditions, the efficiency increases from 80.8% to 86.8% when the TESS is installed. This result can be compared with the installation of a new substation in the middle of the sector. In this case, the efficiency is increased to 91.1%. Nevertheless, as was mentioned in the introduction, the installation of a new substation is not always feasible and can be very expensive. Thus, the implementation of a TESS can be an economical compromise compared with the installation of a new substation. Moreover, it should be noted that a TESS is sized for a given amount of railroad traffic and certain types of trains. Thus, if the traffic grows or if the power of the traction units increases, the TESS must be resized. From this point of view and thanks to the modular solution presented in Figure 3, the TESS is easily expandable.

In Table 6, the sizing of the TESS and the efficiency calculation were carried out for a train power of 4.5 MW. If the power of the trains increases beyond this value, the pantograph voltage drops below 1300 V, and the current in the overhead line increases. Thus, the efficiency decreases dramatically, as shown in Figure 14.

To improve the efficiency of the system, it is, therefore, necessary to limit the voltage drop by increasing the number of elementary blocks in parallel. Thus, according to Table 5, *N* is set to 22 and $V_{threshold}$ is fixed at 1610 V. Considering the railroad traffic in Figure 12, but with $P_{loc}$ = 5 MW, the simulation results show that after 80 min of peak traffic, the batteries are totally discharged. It is, therefore, necessary to further increase the number of elementary blocks in parallel to *N* = 35 while keeping $V_{threshold}$ at 1610 V. In this case, at the end of the peak traffic, the depth of discharge reaches 84.5%. Consequently, 118 min without trains is required to fully recharge the batteries. Compared with the first TESS sizing (*N* = 16), the number of elementary blocks is more than doubled. The cost of the TESS will probably no longer be competitive with respect to the installation of a new substation.

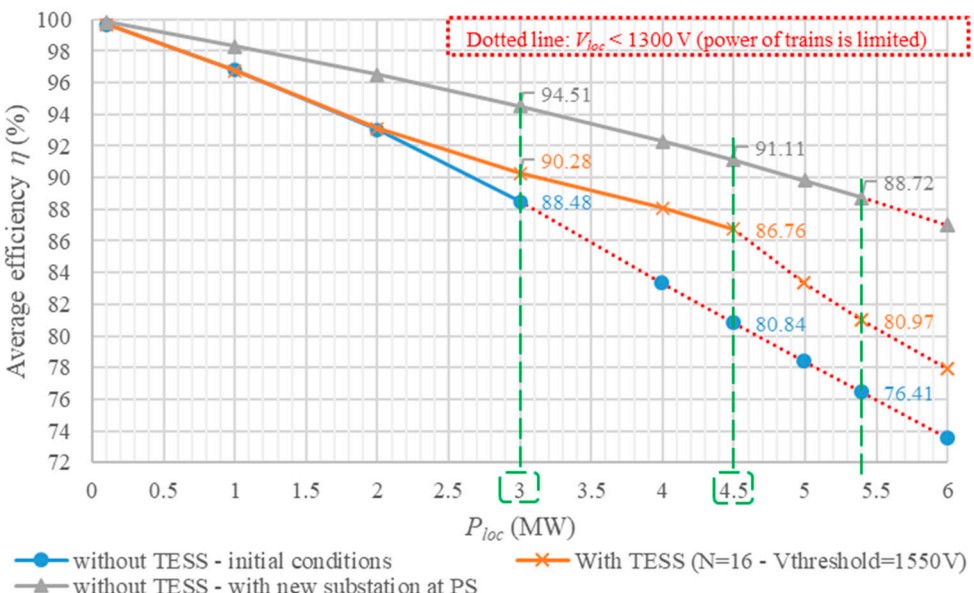

**Figure 14.** Efficiency according to the power of the trains: initial conditions, with TESS ($N = 16 - V_{threshold} = 1550$ V) or with a new substation ($E_0 = 1750$ V; $R_{int} = 25$ mΩ), implemented at PS.

## 4. Test and Experimental Results of a 300 kW/147 kWh Elementary TESS Prototype

As simulation models inevitably have limits, nothing can replace experimentation with a demonstrator. Figure 15 shows the 300 kW/147 kWh prototype of the elementary block of the TESS. The first step, performed in the LAPLACE laboratory test platform, validated the operation of the converter on a resistive load (Figure 15a). In the second phase, the converter, associated with the battery, was implemented in the SCLE-SFE test platform. (Figure 15b). As shown in Figure 15c, a controlled rectifier reproduces the variation in the contact line voltage when trains are circulating. As this thyristor-based rectifier is unidirectional in power, load resistances are connected in parallel to consume the energy during the battery discharge phase. Thus, charging and discharging cycles of the batteries can be performed at full power.

As shown in Figure 16, a human–machine interface (HMI) was developed especially for these platform tests, and it is, in particular, a global synoptic of the system in which many parameters are visible in real-time, such as the positions of the contactors, fault signaling, voltage–current–temperature measurements in the converter, and the entire battery management system (the SoC and voltages–currents–temperatures of each cell).

This HMI allows operation with two different modes: manual and automatic. The manual mode allows the operator to choose the rectifier output voltage, charge/discharge mode, and current level. It is used to debug a system, perform independent tests, and perform selected measurements. The automatic mode allows, as its name suggests, the automation of battery charging and discharging sequences at selected current levels and durations. It counts the number of charge/discharge cycles and performs the chosen number. In addition, the state-of-charge of the battery is monitored, and a choice is made when the SoC reaches its high or low end. On the other hand, balancing the battery cells after a certain number of chosen cycles is possible. This balancing is performed at approximately SoC = 99%, and the converter thus follows a low current set point imposed by the BMS. This automatic mode makes it possible to test the complete system over a long period of time in order to assess its protection and robustness.

Figure 17 shows an example of a typical battery charge/discharge cycle obtained during platform tests of the 300 kW/147 kWh elementary block prototype.

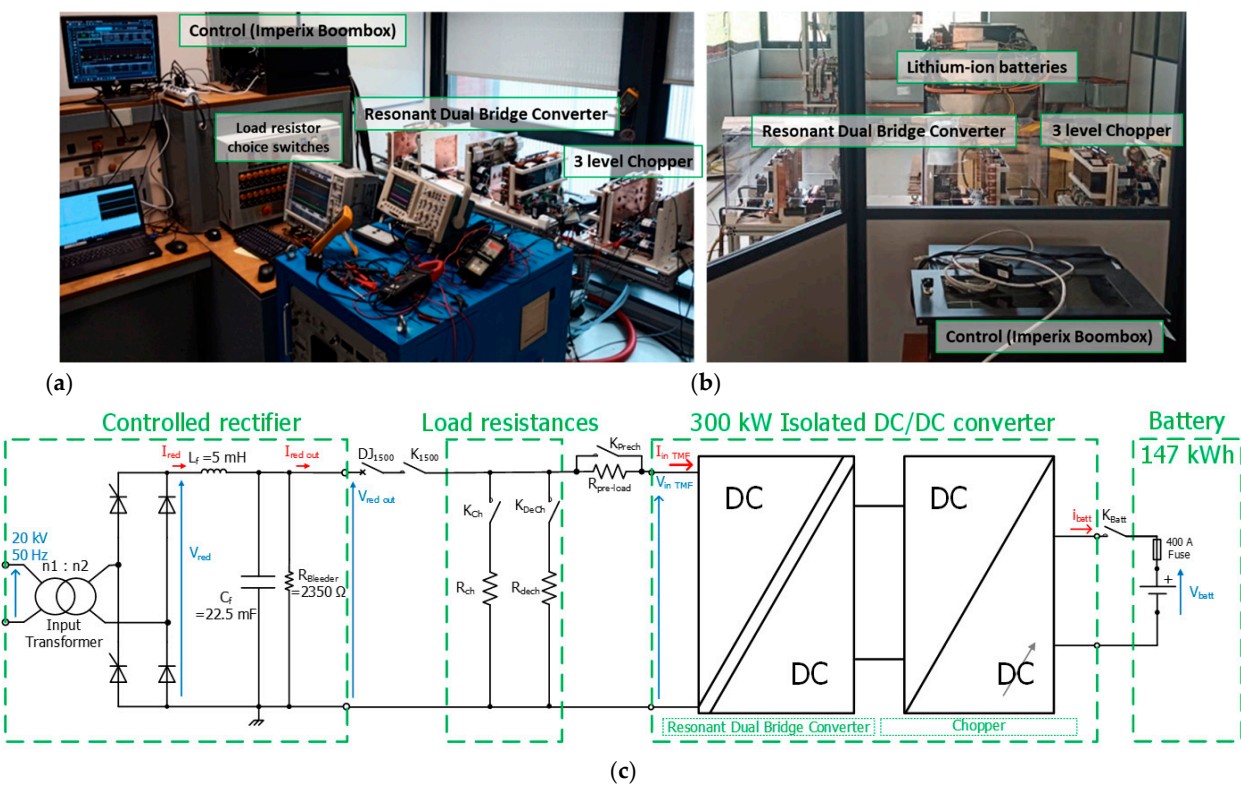

**Figure 15.** Prototype of the 300 kW/147 kWh elementary block: (**a**) picture of the isolated DC/DC converter in the LAPLACE laboratory test platform (resistive load test); (**b**) picture of the isolated DC/DC converter in the SCLE-SFE test platform; and (**c**) electrical diagram of converter under test.

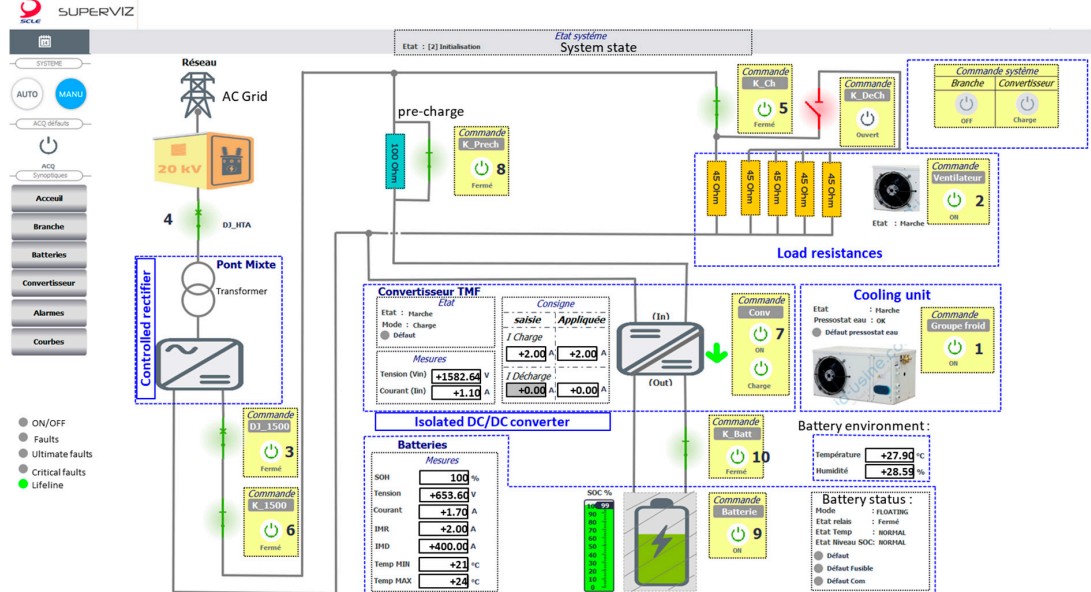

**Figure 16.** Human–machine interface (HMI) display developed especially for platform tests of the 300 kW/147 kWh elementary block prototype.

The converter is powered up by closing the circuit-breaker DJ_1500 and the contactor K_1500. A resistor of 100 Ω allows the charging of all the capacitors in the circuit. Once the voltage has been established in the converters, the pre-charge resistor is short-circuited with the contactor Kprech. Thus, the input voltage of the DC/DC converter is regulated via the thyristor-controlled rectifier (herein, at 1500 V). The contactor K_batt connects the

battery to the converter. The BMS indicates the battery status (for example, herein, the SoC at 90% and the average voltage in the battery at 638 V). The contactors K_Ch and K_DeCh are closed to allow the discharge phase to start with a maximum current setpoint request of 400 A on the HMI. The battery current increases to the setpoint value and remains regulated at this value. During this phase, the input voltage increases because the voltage regulation of the controlled rectifier is slow. The discharge power of the battery is then approximately 255 kW. At t = 2000 ms, the battery current reference returns to 0, and then, the K_DeCh contactor is opened. Afterward, at t = 7000 ms, a charging sequence at a maximum current of 100 A is requested, despite the input voltage decreasing, and the battery current is perfectly controlled. At *t* = 12,000 ms, the charge sequence is stopped. The output voltage of the controlled rectifier increases and returns to 1500 V.

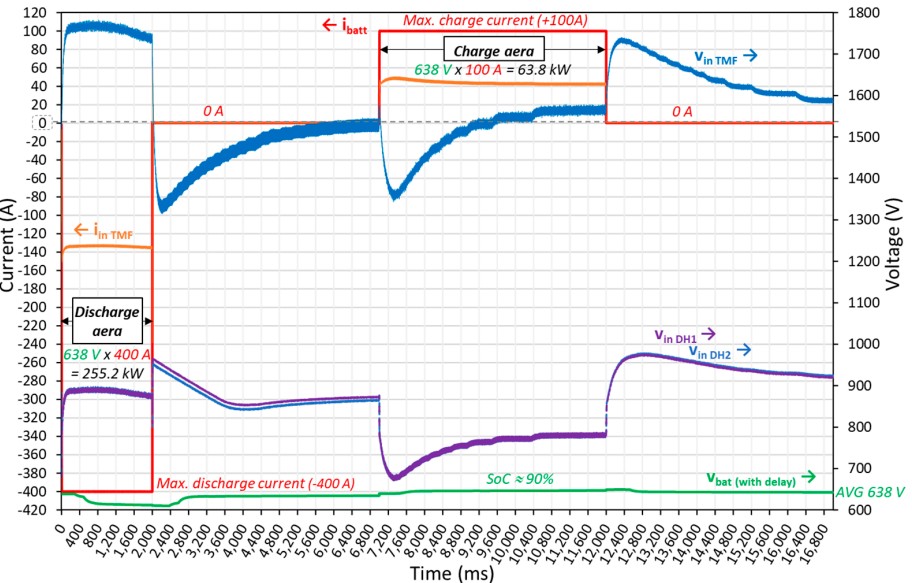

**Figure 17.** Typical battery charge/discharge cycle: platform results with the 300 kW/147 kWh elementary block prototype.

Beyond the accurate regulation of the battery current, this cycle shows correct voltage balancing between the two input capacitors of the chopper (V$_{in\,DH1}$ and V$_{in\,DH2}$).

The instantaneous waveforms for the battery charge and discharge phases are shown in Figure 18.

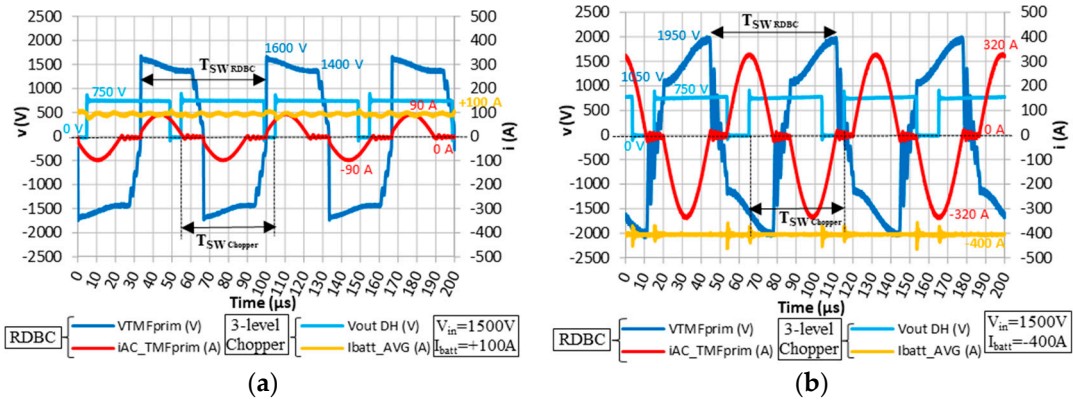

**Figure 18.** Elementary block of 300 kW/147 kWh. Instantaneous waveforms: (**a**) battery charging at maximum current (i$_{batt}$ = 100 A); (**b**) battery discharging at maximum current (i$_{batt}$ = 400 A).

## 5. Conclusions

This paper proposed a new solution for the reinforcement of a DC electrification system with the aim of reducing contact line losses while increasing traffic. The sizing procedure for a TESS as a function of the railroad traffic and the power of the locomotives in circulation was presented. In our case study, considering a scenario of a 1.5 kV double track line with severe railroad traffic, the pantograph voltage drops down to 1000 V and the power drawn by the trains is then reduced. With the installation of a TESS at the PS sized for a total power of 3.84 MW (16 elementary modules in parallel), the pantograph voltage is always greater than 1300 V. Thus, the efficiency of the power system increases from 80.8% to 86.8%. With the implementation of a new substation, the efficiency could be increased to 91.1% but with an installation cost that is certainly higher than that of the TESS.

To validate the topology of the elementary block (300 kW/147 kWh), experiments in a test platform were performed and found to validate the correct operation of the power converter and its control system. For an industrial version sized at 3.6 MW, it is planned to place the isolated DC/DC converters in a 40-foot maritime container, which will also include protection and control devices. Considering a stored energy of approximately 2 MWh, a second 20-foot maritime container including the batteries will be associated with it. Depending on the railroad traffic in the sector, it will be possible to combine several containers of converters and batteries.

Worldwide, 90,000 km of railway routes are electrified with DC. However, these new, isolated DC/DC converters will be essential devices in the evolution of DC electrification systems. Since the galvanic isolation is integrated into the converters, different couplings can be achieved. Thus, the same blocks can be connected in series on the contact line side and can operate on a railway line electrified at 3 kV DC. Beyond energy storage systems, they also could allow the connection of solar power plants to the contact line. Thus, DC railway lines could play the role of energy hubs. All these considerations should encourage railway infrastructure operators to improve the efficiency of their power systems.

**Author Contributions:** Methodology, J.F. and P.L.; writing—review and editing, J.F. and P.L.; Visualization, H.C. All authors have read and agreed to the published version of the manuscript.

**Funding:** This research was funded by the French Agency for Ecological Transition (ADEME) in the frame of the project INSTODRES: INsulated STOrage system for Dc Railway Electrification System.

**Data Availability Statement:** The data presented in this study are available on request from the corresponding author.

**Conflicts of Interest:** The authors declare no conflict of interest.

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
