# Peer review of "Fixed (Trackside) Energy Storage System for DC Electric Railways Based on Full-SiC Isolated DC-DC Converters"

_electronics, doi:10.3390/electronics12071675_

Round 1

Reviewer 1 Report

An article is proposed to the topical issue of our time, dedicated to electric energy storage systems. The application for the railway is considered here. The authors have proposed a new circuit design solution for this field of application. When compared with networks without a ESS, the proposed systems look the most efficient. As it is clear from the paper, the authors suggest installing directly stationary, it may be worth considering the option of a mobile storage device. The conclusions are consistent with the presented evidence, although it would be great to add specific numerical values to the conclusion, how much the proposed system has become better compared to networks without a storage device. In any case, the article can be published after the correction of the conclusion.

Author Response

Point 1: correction of the conclusion (add specific numerical values).

Response 1: Section 3.5 “Analysis of the simulation results” gives elements of comparison between the initial situation of the double-track line sector and the situation with the TESS connected at the PS. Table 6 specifies the energy balance and the efficiency of the power system is presented in Fig. 15 . In addition, to highlight the gain that such a energy storage system can provide, numerical values were added in the conclusion.

Reviewer 2 Report

Dear colleagues, your article has very good quality, and the experimental results are good also. I have a few suggestions. Please comment (if the information is available) about the power density of your solutions compared to the mentioned Japanese solution. Please also include information or directions about how to design the converter, choose devices, the capacitance of capacitors, inductance of inductors, put special attention to the medium frequency transformer, and mention information about the gate drivers. Did you use a commercial solution for the gate drivers? Or you designed it? Fig. 3 shows a coupling inductance between the high-voltage dc-bus and the converter. Please comment on how to choose this inductor.

The English grammar is good. Still, I have included some particular suggestions. Some are not incorrect but uncommon (the English revision is not exhaustive).

On page 1, line 42, it says:

“The power electronics solution developed by Japanese manufacturers do not provide galvanic isolation”

- I suggest changing it for:

“The power electronics solutions developed by Japanese manufacturers do not provide galvanic isolation”

On page 2, line 67, it says:

“the chopper regulates battery current.”

- I suggest changing it for:

“the chopper regulates the battery current.”

On page 2, line 80, it says:

“decreasing manufacturing cost”

- I suggest changing it for:

“decreasing the manufacturing cost”

Author Response

Point 1: Please comment (if the information is available) about the power density of your solutions compared to the mentioned Japanese solution.

Response 1: In the Japanese solutions presented in papers [1] and [2], different battery technologies were tested: nickel hydrogen and lithium-ion. Nevertheless, there is no information on the power density. These solutions are based on classical IGBT choppers operating at a switching frequency of a few kHz, which requires a bulky passive filtering to avoid any interaction with the track circuits. The solution presented in our paper uses a LFP battery and the power converter topology is completely different. A galvanic isolation is provided by a Dual Bridge Converter based on SiC MOSFETs operating at 15 kHz. The passive elements are therefore less voluminous. In addition, it must be considered that the Energy Storage System is installed on the Trackside and the power density is therefore less critical than in the case of an installation on board a locomotive. The converter presented in the paper is a prototype and it is not optimized in terms of power density. Work is currently in progress to achieve an industrial version of this converter with forced-air cooling. We have added in the conclusion a part concerning a future implementation in containers of the power converters and the batteries. Depending on the railroad traffic in the sector, it will be possible to combine several containers of power converters and batteries.

Point 2: Please also include information or directions about how to design the converter, choose devices, the capacitance of capacitors, inductance of inductors, put special attention to the medium frequency transformer, and mention information about the gate drivers. Did you use a commercial solution for the gate drivers? Or you designed it?

Response 2:

In part: 2. "Prototype of the elementary module of the TESS", according to your request, Table 1 and Table 2 were completed with the resonance capacitor values and the parameters of the Medium Frequency Transformer. The sizing and the choice of components were already presented in papers [3] and [4]. The gate-driver of the SiC-MOSFETs was designed at the LAPLACE laboratory. The additionnal reference [14] presents the characteristics and the implementation of this driver.

Point 3: Fig. 3 shows a coupling inductance between the high-voltage dc-bus and the converter. Please comment on how to choose this inductor.

Response 3: To clarify this point, Figure 3 has been modified. For prototype testing, two 1 mH air-core inductors were used at the output of the 3L-chopper. In the industrial version, the chopper will be based on 3.3 kV/750A SiC power modules. Two interleaved chopper legs will be connected in parallel with two coupled inductors. Considering a switching frequency of 7.5 kHz and a current ripple less than 5% of the maximum battery current, the values of inductances Lout1 and Lout2 will be set to 282 µH. As indicated in papers [7], [8], [9], [10], coupled inductors (ICT: InterCell Transformer) will allow an appreciable gain in volume. This new topology will be the subject of a future paper.

Point 3: English revision.

Response 3: Thank you for your corrections of the English language. All modifications are written with the "Track changes" mode of Microsoft Word software.

Round 2

Reviewer 2 Report

My comments have been addressed. Thank you.